# Real-Time Industrial Process Fault Diagnosis Based on Time Delayed Mutual Information Analysis

**Cheng Ji** [ID] **, Fangyuan Ma, Jianhong Wang, Jingde Wang and Wei Sun** *[ID]

College of Chemical Engineering, Beijing University of Chemical Technology, North Third Ring Road 15, Chaoyang District, Beijing 100029, China; 2020400007@mail.buct.edu.cn (C.J.); 2020400053@mail.buct.edu.cn (F.M.); wjhmaster@263.net (J.W.); jingdewang@mail.buct.edu.cn (J.W.)
* Correspondence: sunwei@mail.buct.edu.cn; Tel.: +86-010-6444-5826

**Abstract:** Causal relations among variables may change significantly due to different control strategies and fault types. Off line-based knowledge is not adequate for fault diagnosis, and existing causal models obtained from data driven methods are mostly based on historical data only. However, variable correlation would not remain identical, and could be very different under certain industrial operation conditions. To deal with this problem, a fault diagnosis framework is proposed based on information solely extracted from process data. By this method, mutual information (MI) between each pair of variables is first calculated to obtain thresholds using historical data, as variable correlation under normal conditions is mostly contributed by random noises, which is often neglected in existing causal analysis models. Once a process deviation is detected, each pair of variables with mutual information beyond these thresholds are further investigated by time delayed mutual information (TDMI) analysis using current data, so as to determine the causal logic between them, which is represented as fault propagation paths, can be tracked all the way back to the root cause. The proposed method is first applied to a simulated process and the Tennessee Eastman process. The results show that the difference in variable correlation under diverse operation or control response conditions can be captured in real time, and fault propagation path can be objectively identified, together with the root cause. Then, the method has been successfully applied to a whole year data in an industrial process, which proves the feasibility of industrial application.

**Keywords:** information extraction; fault propagation; time delayed mutual information; industrial application; fault diagnosis

## 1. Introduction

Economic efficiency and process safety are the most important factors in process industry. With the wide application of a distributed control system (DCS), a huge amount of process data can be collected and accumulated, which provides a great possibility for data-driven process monitoring [1]. In recent years, a significant number of advanced monitoring methods have been proposed for early fault detection and diagnosis.

In DCS, each measurement is preset an operating range. An alarm will be triggered if any variable exceeds the predefined operating limit. However, unnecessary alarms can be quite overwhelming if there is no effective fault detection and diagnosis system [2], which requires massive process knowledge and operation experience. To avoid this problem, many data-driven fault diagnosis methods are further proposed based on data collected from DCS. Among these methods, multivariate projection methods are the most commonly used by extracting key features of given process data and calculating the variable contribution with respect to the occurring fault [3]. The contribution of each variable is plotted in a histogram for comparative analysis, and the variable with the highest contribution is considered as the root cause of the fault [4]. The contribution plots are easy to calculate, but, for modern industrial chemical processes, variables interact with each other under the influence of equipment, stream loop, and control strategy. Once a fault occurs, the

fast propagation of the contribution from root variable to other variables could be difficult to recognize [5]. For this purpose, a causal reasoning model has been established and further combined with the contribution plots to solve this problem. If causal reasoning among variables could be obtained, the variables that have significant contributions could be arranged in a network diagram, from which the fault propagation could be obtained and then the real root cause could be located.

There are two ways to extract the causal logic among process variables, i.e., by knowledge and by data. For knowledge-based methods, intensive process knowledge and operational experience are required [6]. With the increasing scale and complexity of chemical equipment and process topology, enough process knowledge is difficult to obtain and hard to update with an ever-changing operating condition, equipment aging and switch of control strategies. Motivated by this consideration, data driven methods are preferred with no prior process knowledge required.

Causal logic among process variables can also be described as correlation with a proper time lag between each pair of variables. One of the simplest and widely used methods is a Pearson correlation coefficient, but it is a linear correlation analysis, cannot be applied to extract nonlinear correlation, and also no time lag is considered. Several data driven causal reasoning methods like Granger causality [7] and transfer entropy [8] have been applied to build causal networks of process variables for root cause analysis. The Granger causality was first proposed as an economics theory to identify causal interactions and played an important role in explaining economic phenomena, which makes it more and more popular and can be applied to physics [9], bioinformatics [10], neuroscience [11], and many other research fields. However, the Granger causality is established on partial least squares, which is a linear regression method too. For most systems in chemical industrial processes, the correlation among process variables is highly nonlinear. The application of Granger causality in these nonlinear systems is inadequate [12]. To extract nonlinear correlation, transfer entropy is proposed based on information theory by calculating how much the uncertainty of one signal can be reduced if another signal is known [13]. Transfer entropy is derived from information entropy (IE). Its mathematical calculation is established by estimating probability distribution; therefore, the nonlinear correlation between variables is well considered [14]. It has been applied to determine the correlation among process variables in many nonlinear systems and shows a good performance on root cause analysis on faults defined in the Tennessee Eastman process (TEP) [15].

Up to date, most causal reasoning methods are established based on data from normal operation conditions, which are then applied to illustrate a fault propagation path by a signed directed graph, once a fault is detected by a contribution plot. Due to a complex process topology of a chemical process, there are massive control loops to ensure stable operation of the system. When a fault occurs, the causal logic among process variables may change with the response of the control strategies, making the diagnosis results inconsistent with practical situations [16]. In this work, the causal logic among process variables is obtained by analyzing mutual information (MI) from both historical data and real-time data, and then applied to identify fault propagation path and root cause. Better results are obtained compared with process knowledge-based methods and traditional data driven methods.

The following parts of this paper are arranged as follows: in Section 2, the IE and MI are briefly introduced. The establishment of the digraph model based on time delayed mutual information (TDMI) is also detailed. In Section 3, the procedure of the proposed fault diagnosis method and the selection of parameters are described, followed by a simple nonlinear simulation example. In Section 4, the proposed method is applied to two industrial processes, TEP, and an industrial reforming process. The fault isolation results are shown and discussed. In Section 5, the paper is concluded.

The abbreviations used in this work are summarized and interpreted as follows: MI represents mutual information; TDMI represents time delayed mutual information;

DCS represents a distributed control system; IE represents information entropy; and TEP represents the Tennessee Eastman process.

## 2. Preliminaries

In this section, the IE, MI, kernel density estimation method, and digraph model are introduced as the basis of the proposed method.

### 2.1. Information Entropy

IE is proposed by Shannon based on the concept of thermodynamics entropy, representing the uncertainty of the variable [17]. Given a random signal $X$ $(x_1, x_2, \ldots, x_n)$, its IE can be calculated as follows:

$$H(X) = -\sum_{i=1}^{n} p(x_i) \log(p(x_i)) \tag{1}$$

where $p(x_i)$ is the probability distribution of the sample $x_i$, with assumption that the IE is only related to the probability distribution of the variable. The probability distribution of the variable when its IE reaches the maximum value can be obtained by the Lagrange multiplier method:

$$F(p(x_1), \ldots, p(x_n)) = -\sum_{i=1}^{n} p(x_i) \log(p(x_i)) + \lambda(\sum_{i=1}^{n} p(x_i) - 1) \tag{2}$$

$$\frac{dF(p(x_1), p(x_2), \ldots, p(x_n))}{dp(x_i)} = \lambda - 1 - \ln(p(x_i)) \tag{3}$$

where $\lambda$ is the Lagrange multiplier; in order to get the maximum value, the derivative in Equation (3) should be equal to 0. Then, the probability distribution $p(x)$ can be solved as follows:

$$p(x_i) = e^{\lambda - 1} \tag{4}$$

$$\sum_{i=1}^{n} p(x_i) = ne^{\lambda - 1} = 1 \tag{5}$$

$$p(x_i) = \frac{1}{n} \tag{6}$$

It can be concluded that the more uniform the probability distribution of the variable is, the less information the data contains, which means that the uncertainty of the variable is high. In contrast, if $p(x_i) = 1$, variable $x$ is completely determined, the IE value will be reduced to 0 according to Equation (1). In information theory, IE is extended to multi-variables to measure the uncertainty of a system.

### 2.2. Mutual Information

MI is a measurement for the correlation between two measurements from the aspect of IE. Given two random variables $x$ and $y$, MI measures the effect on the uncertainty of $x$ if $y$ is given [18]:

$$
\begin{aligned}
I(X, Y) &= H(X) - H(X|Y) \\
&= -\sum_x p(x) \log(p(x)) + \sum_x \sum_y p(x, y) \log(p(x|y)) \\
&= -\sum_x \sum_y p(x, y) \log(p(x)) + \sum_x \sum_y p(x, y) \log(p(x|y)) \\
&= \sum_x \sum_y p(x, y) \log(\frac{p(x,y)}{p(x)p(y)})
\end{aligned}
\tag{7}
$$

where $p(x, y)$ is joint probability distribution. According to Equation (7), if $x$ and $y$ are independent, $p(x, y)$ is equal to 0, and the MI will be 0, while, if there is a strong correlation between $x$ and $y$, the MI will be large.

It can be known from the definition that the correlation measured by MI is only related to the probability distribution of the variables. Therefore, the quantification of MI is not limited to linear correlation between variables. Compared with commonly used Pearson correlation, MI is obtained based on both linear and nonlinear relationships between measurements. Take the following relationship as an example:

$$\begin{aligned} x &= \sin(t) \\ y &= \cos(t) \\ t &= 1, 2, 3, 4, \ldots, 1000 \end{aligned} \tag{8}$$

As shown in Equation (8), there is a trigonometric relationship between the variables $x$ and $y$. The Pearson correlation coefficient between $x$ and $y$ is 0.0009032, which means there is almost no linear correlation between them, while the normalized MI is 0.4606, indicating that there is a certain correlation between the variables $x$ and $y$.

Because of the symmetry in the calculation of MI, it can only reflect the degree of correlation between variables, but the chronological order of information between variable deviations is not available. To overcome this limitation, a time lag parameter can be introduced in the calculation to identify chronological order between variable deviations [19]. The MI with a time lag can be defined as follows:

$$I(X, Y, \tau) = \sum_{x_t} \sum_{y_{t+\tau}} p(x_t, y_{t+\tau}) \log\left(\frac{p(x_t, y_{t+\tau})}{p(x_t)p(y_{t+\tau})}\right) \tag{9}$$

where $p(x, y + \tau)$ is the joint probability distribution of $X = x_t$, $Y = y_{t+\tau}$, and $\tau$ is the time lag parameter. The parameter $\tau$ is determined when the MI reaches its peak. A positive $\tau$ indicates that $x$ has a maximum correlation with future $y$, which means that $x$ changes before $y$, while negative $\tau$ indicates that $x$ changes after $y$. A simple nonlinear example is used to demonstrate this point:

$$y(t) = 0.5x(t - \tau) - 0.2x(t - \tau)^2 + s_1 \tag{10}$$

where $x(t)$ is a random variable follows a uniform distribution between 0 and 1, and $s_1$ is a random noise that follows a normal distribution with mean 0 and variance 0.02. Another variable $y(t)$ has a nonlinear relationship with $x(t - \tau)$, and $\tau$ is set to several different values ($-20, -10, 0, 5, 15, 25$) to test the performance of TDMI. The MI of $x$ and $y$ with a time lag between $-30$ and 30 is shown in Figure 1, and it can be seen that the MI reaches its peak when the time lag equals to the preset $\tau$ in all the six situations. Due to the causal feature of industrial process, only variables changing earlier can possibly be the cause for other variables changing later; therefore, the TDMI can be applied to detect and analyze the causal relationship in nonlinear systems.

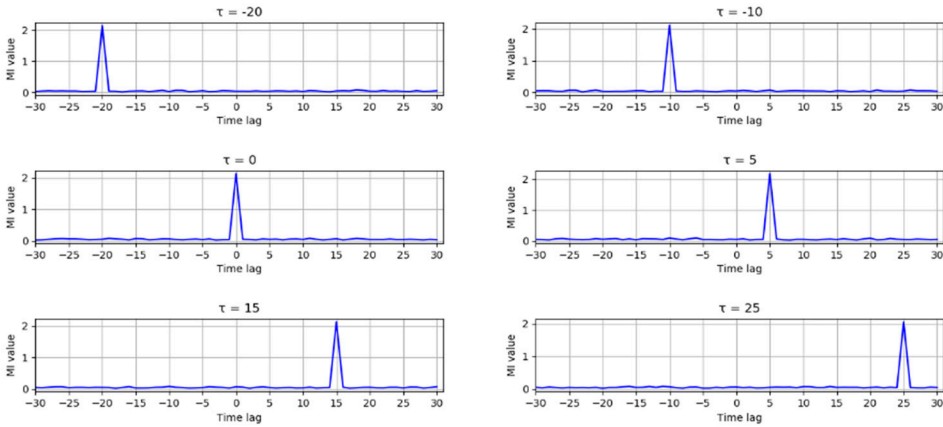

**Figure 1.** Time delay detection results.

*2.3. Kernel Density Estimation Method*

Because the actual probability distribution of industrial data is unknown, the calculation of IE and MI mainly lies in the estimation of probability distribution. In this work, non-parametric probability density estimation methods are used, as it can be applied to estimate any form of probability distribution without assumptions. Among the commonly used non-parametric estimation methods, the kernel density estimation method is selected for its good performance on probability density estimation of a small number of samples [20].

In the kernel density estimation method, a kernel function *K* is applied to obtain a probability density function at every sample point, and all of these probability density functions are summed and averaged to obtain the estimate *p(x)*:

$$p(x) = \frac{1}{nd} \sum_{i=1}^{n} K(\frac{x - x_i}{d})$$ (11)

where *n* is the number of samples and *d* is the window width which is adjusted with *n* and the standard deviation of the variable *x* to give a good estimation of the actual probability density function. Generally, a Gaussian function is used as the kernel function, and the optimal width is:

$$d = 1.06\sigma n^{-1/5}$$ (12)

where *σ* is the standard deviation of the variable. Equation (12) is derived by minimizing the mean integrated squared error function [21].

*2.4. Digraph Model*

The identification of a cause and effect relationship in a system plays an important role in root cause diagnosis. The digraph model is a widely used intuitive tool to display the causal relationship. Generally, a digraph model can be described as *G* (*N*, *A*), where *N* is the nodes of the variables, and A is the directed arcs which connects cause nodes to consequent nodes.

A signed directed graph model is the most commonly used digraph model for fault diagnosis, in which a sign either positive or negative is set in each arc to represent whether the cause and consequent node change in the same direction or an opposite direction [22]. Arcs and signs are usually established from process knowledge or expert experience of a process. Once a fault occurs, a sign in each node is set by comparing its value with a normal range determined previously. If the node value is higher than the normal range, the sign is set to positive. If the node value is lower than the normal range, the sign is set to negative. The sign is set to 0 if the node value is within the normal range. The sign of an arc in a consistent path is defined as positive, if the signs in both nodes are same, which means the nodes at both end of the arc changes in the same direction. Otherwise, the sign of the arc is defined as negative, if the signs in both nodes are opposite. Therefore, the fault propagation can be obtained by finding the consistent path in a signed directed graph model.

However, the challenge of the signed directed graph model is that it is hard to include enough process knowledge required in a complex industrial process. In this work, the digraph model is established by a data driven method. The nodes in the proposed model are obtained by finding the variable nodes whose IE is out of the normal range defined by previously calculated IE. The direction of the arcs is determined from the TDMI between two nodes. With the signed directed graph, the propagation path can be obtained only from process data, which provide a more objective information for fault diagnosis.

## 3. Fault Diagnosis Method with Information Solely Extracted from Process Data

In this section, IE and TDMI introduced in a previous section are used to propose a strategy for root cause diagnosis with both historical data and real-time data. The entire process of the proposed method and the details of the diagnosis strategy is described below.

### 3.1. Procedures for Process Fault Diagnosis

The basic idea of proposed diagnosis method is to estimate the probability density of process variables and then calculate their MI with different time lags. Generally, MI between variables, which are affected by faulty deviation, shows different characteristics from that under normal operating conditions. It can reach its maximum with a proper time lag, and the time dependency between these variables is therefore determined, which is usually depicted by a directed graph for fault propagation analysis.

As mentioned before, it requires significant computation to calculate the MI between every pair of variables if kernel density estimation is employed, as only part of the process variables will be affected under a faulty condition. These variables can be selected by their information entropies first. IE can be considered as the uncertainty or the amount of information contained in a variable. Given a random variable $x$ in the industrial process, $x$ generally fluctuates around its set point under the influence of noise under normal operation conditions. In this case, less information corresponds to a higher IE. Once a fault occurs in this process, the value of $x$ will deviate from its set point, if it is affected. The amount of information contained in $x$ will increase accordingly; in the meantime, its IE will decrease. On the other hand, the distribution of $x$ will change under a faulty condition, and further affect its IE, which is only determined by its probability distribution according to its definition. Therefore, the variable is selected if its IE is out of the range obtained under normal operation conditions.

Once affected variables are identified, the time dependency between these variables can be determined by TDMI. Because there is no upper bound in MI, it is difficult to determine whether the variable correlation is significant based on MI values alone. When a fault occurs, MI shows different characteristics from that under normal operation conditions. When a system is under normal conditions, most process variables fluctuate randomly near their set value, and therefore the actual correlation between variables cannot be revealed. In this case, most MI is contributed by random noises. Once a fault occurs, corresponding changes will happen in certain variables. If there is causal relationship between these variables, the MI between them will increase with fault propagation and exceed the range determined under normal operation conditions. Next, a simple nonlinear example is applied to prove this point:

$$\begin{aligned} x_t &= 2.8 + e_1 \\ y_t &= 0.1{x_t}^2 + e_2 \end{aligned} \tag{13}$$

where $x$, $y$ is random variables with a nonlinear correlation, $e_1$, $e_2$ is random noises that follows a normal distribution with variance 0.01 and 0.02. A set of 1000 samples is simulated and a ramp fault with a slope of 0.005 is introduced in $x$ from 500 to 600 samples. The MI of $x$ and $y$ is calculated with a moving window and compared under different conditions to prove the correlation between $x$ and $y$. The distribution of MI under normal operation conditions is shown in Figure 2a.

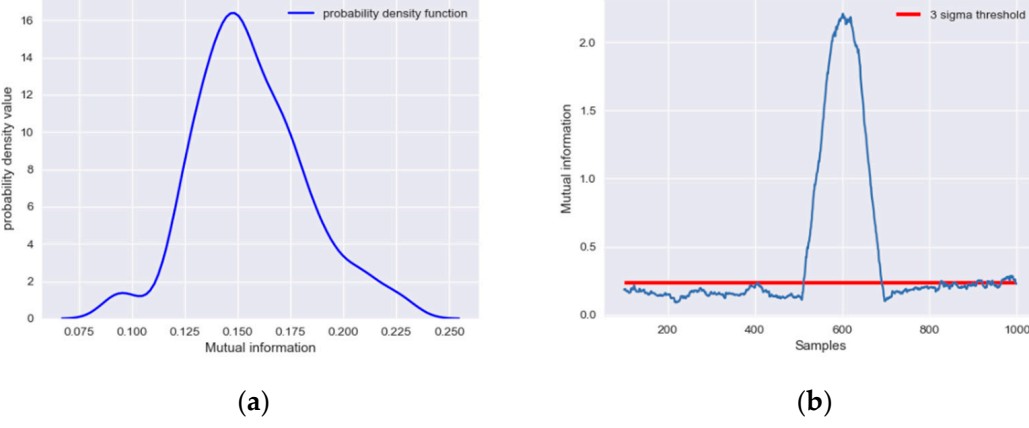

|                  |                  |
|:----------------:|:----------------:|
| **(a)**          | **(b)**          |

**Figure 2.** (**a**) Probability distribution of MI under normal conditions; (**b**) MI in each moving window.

It can be seen that the MI follows a normal distribution with a relatively small mean. A three-sigma threshold is chosen here to define the significance level:

$$s(x,y) = \frac{I(x,y) - \mu}{\sigma} > 3 \tag{14}$$

where $s(x, y)$ represents that the correlation between $x$ and $y$ is significant, $I(x, y)$ is the MI, and $\mu$ and $\sigma$ are mean and standard deviation of MI calculated under normal operation conditions.

The MI of $x$ and $y$ from all the moving windows is shown in Figure 2b. After the 500th sample, abnormal data are contained in the calculation windows of MI; therefore, the MI of $x$ and $y$ increases significantly and exceeds the threshold determined under normal conditions, indicating that there is a significant correlation between $x$ and $y$.

### 3.2. The Implementation of the Proposed Fault Diagnosis Framework

The flowchart of the proposed fault diagnosis method is shown in Figure 3, and its implementation includes following two parts:

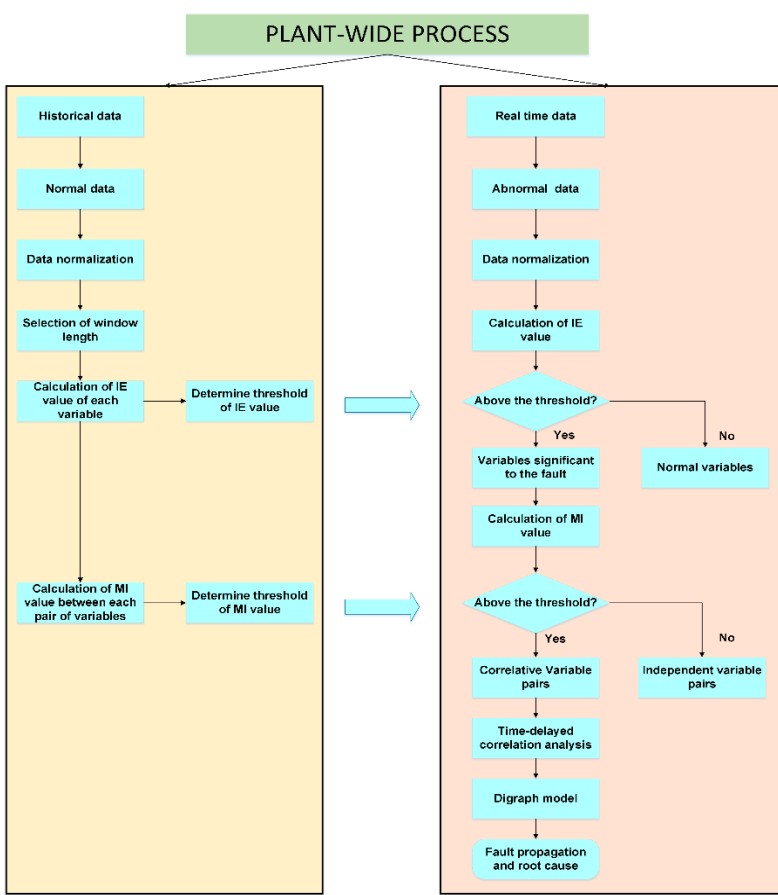

**Figure 3.** The framework of the proposed fault diagnosis method.

Offline parameters selection:
(1) Select data under normal operation conditions from historical data.
(2) Normalize the data and choose a suitable window length for the calculation of IE and MI.
(3) Calculate IE with a moving window based on kernel density estimation and determine the threshold of each variable.
(4) Calculate the MI of each pair of variables with the moving window and determine the threshold.

Online fault diagnosis:

(1) Collect real time data, once a fault is detected. These data are usually referred to as abnormal data.
(2) Calculate the IE of each variables using abnormal data and compare it with the thresholds determined previously. Variables that exceed the threshold are selected as the fault nodes in the directed digraph.
(3) Calculate MI of each pair of variables selected in the last step and compare it with the thresholds obtained offline. Variables that exceed the threshold indicate a significant correlation between them, and the nodes are connected with directed arcs in the digraph.
(4) Calculate the TDMI between correlated variables obtained in the last step to determine the direction of arcs.
(5) Isolate fault and analyze fault propagation path in the digraph. Root node can be regarded as the source of the fault, and child nodes are regarded as the consequent caused by the fault.

*3.3. A Root Cause Identification in a Simulated Example*

A nonlinear process contains two variables defined in following equations:

$$x_t = 2.8 + e_1$$
$$y_t = x_{(t-3)} * (1.8 - 0.5 * x_{(t-3)}) + e_2 \tag{15}$$

where $x_t$ is a constant random variable with a random noise $e_1$ that follows a normal distribution with variance 0.01, $y_t$ is another random variable that is caused by $x_t$. Time lag of the correlation between $x_t$ and $y_t$ is set to 3. $e_2$ is a random noise that follows a normal distribution with variance 0.02. A set of 2000 samples is simulated, and a ramp fault with the slope of 0.005 is introduced to $x_t$ from the 1500th to 1700th sample. The variables are illustrated in Figure 4. The first 1000 samples are selected as normal data to determine thresholds of IE and MI.

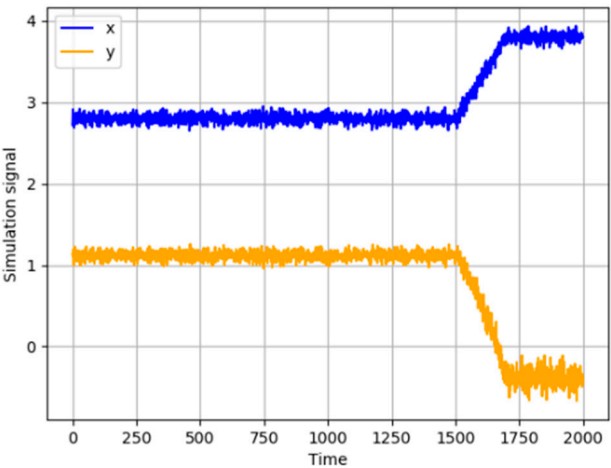

**Figure 4.** Original data of the simulation signal.

As discussed in the previous section, distribution characteristics of correlation between process variables are obtained by calculating MI with a moving window. The length of the moving window has a great influence on the estimation of probability density and further affects the calculation results of MI. For this concern, different window sizes need to be investigated. The MI calculated at window length from 30 to 1000 is shown in Figure 5.

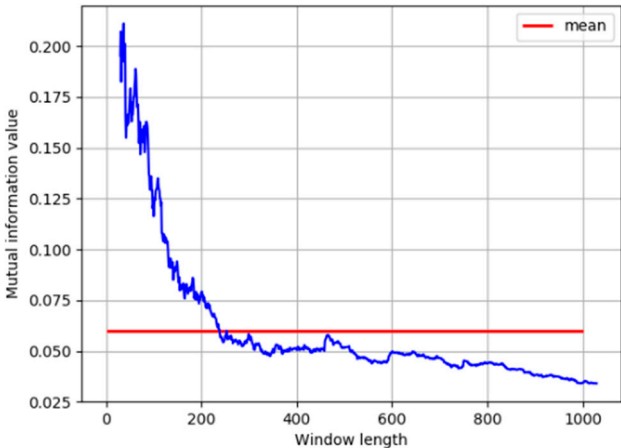

**Figure 5.** MI in different window lengths.

It can be seen that, when the window is short, the MI results are unreliable due to the insufficient number of samples in kernel density estimation. When the window reaches a certain size, the MI remains stable as the window length increases, and is close to its average value under all window sizes. Therefore, the window size is selected as 240.

Then, thresholds of IE and MI are determined with the first 1000 normal data. The IE of $x$ and $y$ under normal operating conditions and the distribution of MI of $x$ and $y$ are shown in Figure 6a,b. It can be obtained that the MI follows a normal distribution with mean 0.07728 and standard deviation 0.01315. As discussed before, a three-sigma threshold is selected as 0.15617, and the thresholds of IE are selected in the same way.

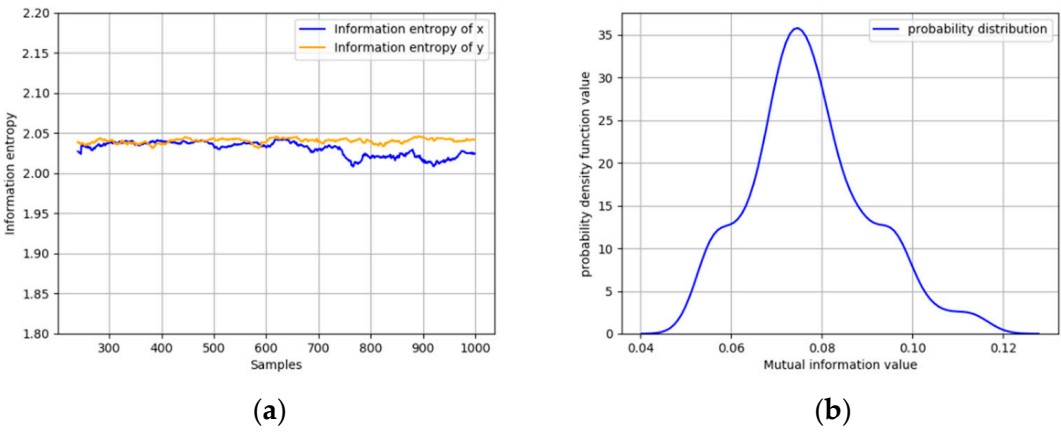

(**a**)                      (**b**)

**Figure 6.** (**a**) IE calculated under normal conditions; (**b**) distribution of MI under normal conditions.

When the ramp fault occurs, the IE of $x$ and $y$ are calculated. As shown in Figure 7a, both the IE of $x$ and $y$ decrease significantly and fall below the thresholds, indicating that the distributions of $x$ and $y$ both deviate from the normal operating condition. In order to diagnose the root cause of this fault, the MI between $x$ and $y$ is calculated and compared with its threshold. It can be seen from Figure 7b that the MI between $x$ and $y$ exceeds its threshold when the fault occurs, which means that the fault could be propagated between $x$ and $y$. The direction of the interaction between $x$ and $y$ is identified based on the TDMI in Figure 8. The time lag is introduced in $y$, and it is obvious that the MI reaches its peak when the time lag is equal to 3, which means $y$ is caused by $x$. Therefore, the fault propagation path can be easily determined as from $x$ to $y$. The results show that the TDMI based method successfully identifies the root cause, together with its propagation path. Considering more numbers of process variables and more complicated variable correlation in industrial

processes, the fault diagnosis results are displayed in the form of the directed graph in the following case studies.

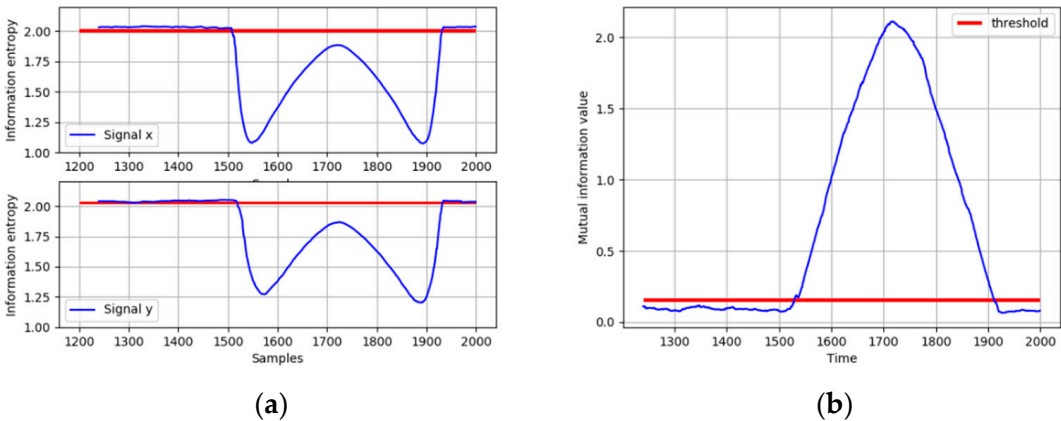

(**a**)　　　　　　　　　　　　　　　　　(**b**)

**Figure 7.** (**a**) IE calculated under abnormal conditions; (**b**) MI calculated under abnormal conditions.

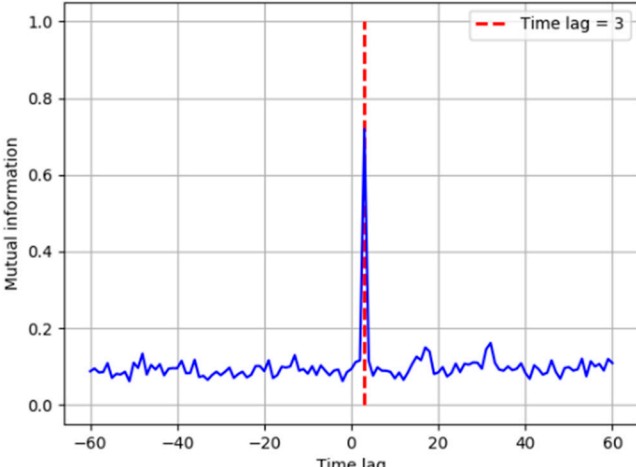

**Figure 8.** MI with different time lags.

## 4. Case Studies

### 4.1. Tennessee Eastman Process

In this section, TEP is applied as a case study to test the proposed method. TEP is a widely used chemical process simulated from a real industrial process plant by the Eastman Chemical Company [23,24]. The process flow chart of TEP is shown in Figure 9. Feed streams flow through a reactor, condenser, vapor/liquid separator, compressor, and a stripper, respectively, to implement the generation and separation of products. There are totally 52 variables in the process, including 11 manipulated variables, 22 continuous measured variables, and 19 composition variables. In this work, composition variables are not considered because the sampling period of them is long, and their impact under normal operating conditions is negligible. Variables selected in this work are shown in Table 1. In addition, a set of 21 preset faults are simulated in TEP for testing and comparing various new algorithms. The description of these faults is shown in Table 2. The proposed method is applied to a normal data set and all 21 faulty data sets to diagnose the root cause. Each data set contains 960 samples with a sampling period of 3 min and all 21 faults are introduced at the 160th sample point [25].

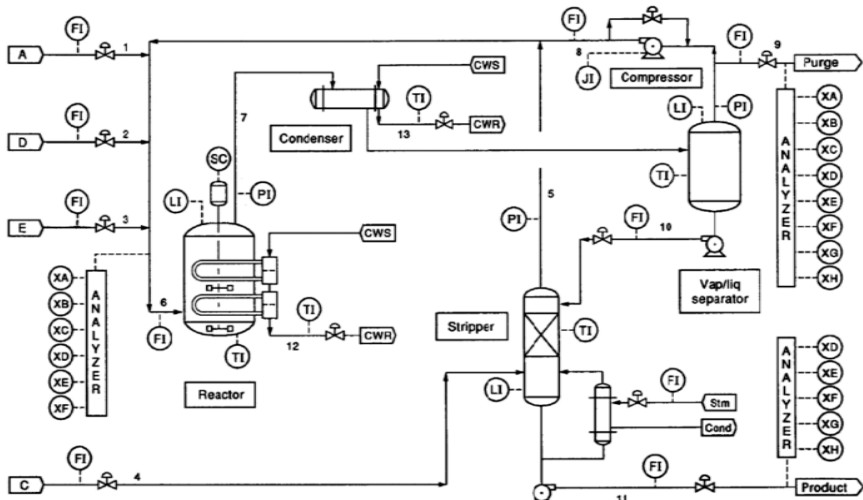

**Figure 9.** Tennessee Eastman process flow chart. Reproduced with permission from [Downs, J.; Vogel, E.], [Computers & Chemical Engineering]; published by [Elsevier], 1993.

**Table 1.** The variables in the Tennessee Eastman process.

| Variable | Description | Variable | Description |
|---|---|---|---|
| $F_1$ | A feed (stream 1) | $T_{18}$ | Stripper temperature |
| $F_2$ | D feed (stream 2) | $F_{19}$ | Stripper steam flow |
| $F_3$ | E feed (stream 3) | $C_{20}$ | Compressor work |
| $F_4$ | A and C feed (stream 4) | $T_{21}$ | Reactor cooling water outlet temperature |
| $F_5$ | Recycle flow (stream 8) | $T_{22}$ | Separator cooling water outlet temperature |
| $F_6$ | Reactor feed rate (stream 6) | $V_{23}$ | D feed flow (stream 2) |
| $P_7$ | Reactor pressure | $V_{24}$ | E feed flow (stream 3) |
| $L_8$ | Reactor level | $V_{25}$ | A feed flow (stream 1) |
| $T_9$ | Reactor temperature | $V_{26}$ | A and C feed flow (stream 4) |
| $F_{10}$ | Purge rate (stream 9) | $V_{27}$ | Compressor recycle valve |
| $T_{11}$ | Product separator temperature | $V_{28}$ | Purge valve (stream 9) |
| $L_{12}$ | Product separator level | $V_{29}$ | Separator pot liquid flow (stream 10) |
| $P_{13}$ | Product separator pressure | $V_{30}$ | Stripper liquid prod flow (stream 11) |
| $F_{14}$ | Product separator underflow (stream 10) | $V_{31}$ | Stripper steam valve |
| $L_{15}$ | Stripper level | $V_{32}$ | Reactor cooling water flow |
| $P_{16}$ | Stripper pressure | $V_{33}$ | Condenser cooling water flow |
| $F_{17}$ | Stripper underflow (stream 11) | | |

**Table 2.** The process faults in the Tennessee Eastman process.

| No. | Fault Description | Fault Type |
|---|---|---|
| 1 | A/C feed ratio, B composition constant (stream 4) | Step |
| 2 | B composition, A/C ratio constant (stream 4) | Step |
| 3 | D feed temperature (stream 2) | Step |
| 4 | Reactor cooling water inlet temperature | Step |
| 5 | Condenser cooling water inlet temperature | Step |
| 6 | A feed loss (stream 1) | Step |
| 7 | C header pressure loss-reduced availability (stream 4) | Step |
| 8 | A, B, C feed composition (stream 4) | Random variation |
| 9 | D feed temperature (stream 2) | Random variation |
| 10 | C feed temperature (stream 4) | Random variation |
| 11 | Reactor cooling water inlet temperature | Random variation |
| 12 | Condenser cooling water inlet temperature | Random variation |
| 13 | Reaction kinetics | Slow drift |
| 14 | Reactor cooling water valve | Sticking |
| 15 | Condenser cooling water valve | Sticking |
| 16 | Unknown | - |
| 17 | Unknown | - |
| 18 | Unknown | - |
| 19 | Unknown | - |
| 20 | Unknown | - |
| 21 | The valve for stream 4 | Constant position |

The normal data set is first used to determine threshold of IE and MI value. The length of the moving windows is selected as 100 based on the principle mentioned previously. Therefore, a total of 860 IE value as well as 860 MI value for each pair of variables can be obtained; then, the thresholds can be determined. Fault diagnosis results of faults in TEP determined by the proposed method are shown in Table 3. The implementation of root cause diagnosis of fault 1 (A and C feed ratio failures in stream 4) is analyzed based on the proposed method. In the results of this work, the red node represents the cause node, the blue node represents consequent node, and the green node represents the node that is ultimately affected.

**Table 3.** Fault diagnosis results in TEP determined by the proposed method.

| Fault No. | Fault Diagnosis | Fault Analysis |
|---|---|---|
| 1 | (16,20,27)→25→(7,13)→1 | Pressure disturbance in stripper (from stream 4) |
| 2 | (28,10) | Purge valve varies with the disturbance in B component |
| 4 | (9,32) | Cooling water temperature disturbance in reactor |
| 5 | 22→(11,13,16) | Variation of cooling water temperature |
| 6 | 1→(7,13,16)→25 | Variation of flow in A feed (stream 1) |
| 7 | 4,9,16,26→23 | Pressure disturbance in stripper (stream 4) |
| 10 | 18 | Temperature disturbance in stripper (from stream 4) |
| 11 | (9,32) | Cooling water temperature disturbance in reactor |
| 12 | 22→11 | Variation of cooling water temperature |
| 14 | 32→21→9 | Reactor cooling water valve failure |
| 17 | 21→32 | Reactor cooling water outlet temperature disturbance |
| 18 | 22 | Separator cooling water outlet temperature disturbance |
| 20 | 27→18 | Pressure disturbance in stripper |
| 21 | 26 | A and C feed flow (stream 4) valve failure |

### 4.1.1. Fault Propagation Analysis in the Tennessee Eastman Process

According to the description of fault 1, the root cause is the step variation of A/C feed ratio, but a variable directly related to the fault is not monitored [26]. In some studies, the root cause is located to the A and C feed, which is not consistent with the actual situation because the total flow of the A and C feed in process data remains stable when this fault is detected [27]. In Kim's studies, the root cause is determined to be the variables that are closely related to and are greatly affected by the fault [28]. The route of fault propagation from his work is determined as C20→P16→P13→P7→V28→V25. In his explanation, the fault is related to the change of pressure and the compressor work is adjusted to mitigate this change, which means that the fault propagation from C20 to P16 is questionable. The compressor is not the nearest location where this fault is introduced either. Though the A/C feed ratio is not monitored, it can be noted from the process flow chart that the A/C feed is from stream 4 and can be transported to the stripper. Therefore, the fault is introduced to stream 4, and the closest related equipment is the stripper.

As shown in Figure 10, the IE value of each variable is calculated and compared with its threshold determined from training data. It can be observed that there are seven variables in total that exceed the threshold, and the fault propagation among these variables is shown in Figure 11. The root nodes are stripper pressure, compressor work, and compressor recycle valve. Based on the root nodes, it can be concluded that there are two possible reasons for this fault. One is the fault that is caused by the pressure, and the compressor recycle valve is quickly adjusted to mitigate this change. The other is that the fault occurs in the compressor recycle valve, resulting in changes in the pressure of each equipment. According to the subsequent fault propagation, the fault is successively propagated to the A feed flow, the reactor pressure, product separator pressure, and the A feed. Once the fault occurs in the compressor, the fault should be propagated to the stripper pressure as well as reactor pressure and separator pressure, and the A feed flow will not be influenced along, therefore, the fault can be directly located in the stripper pressure. At the same time, the stripper pressure cannot be the real root cause as a state variable, and the fault is propagated to the A feed flow at last, so the fault could be caused by the feed rate near the stripper. From the process flow chart, the C feed flow is directly delivered to the stripper.

Because the A and C feed is not in the fault propagation path, the fault may occur in the A/C feed ratio (stream 4), which is consistent with the fault description. As this variable is not measured in this system, the root cause is only located to the stripper pressure. The stripper is the nearest equipment from where the fault occurs, and it is directed affected by the fault location stream 4.

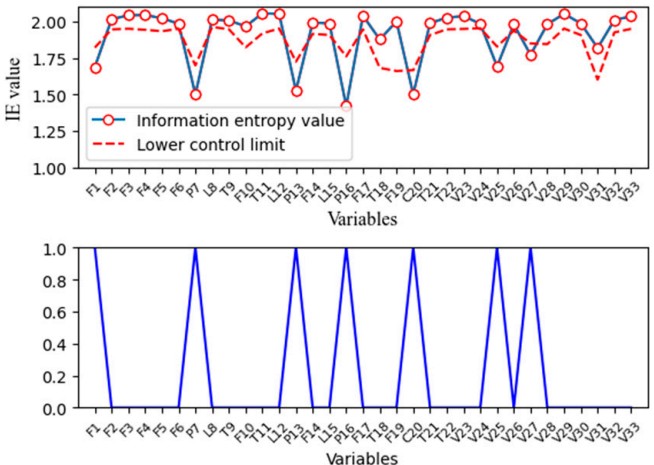

**Figure 10.** IE value of each variable in fault 1.

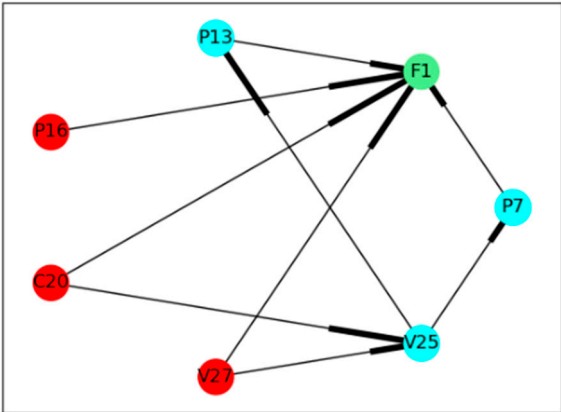

**Figure 11.** Fault propagation digraph in fault 1.

### 4.1.2. The Difference between the Correlation of Variables under Normal Operation and Abnormal Operation

As mentioned, correlation among process variables under normal operation and abnormal operation can be different or even completely reversed due to the existence of control systems, which can be well proved by the fault diagnosis results of faults 11, 14, and 17. IE values of each variable in fault 14 and fault 17 are shown in Figure 12a,b. Fault propagation results of fault 14 and fault 17 are shown in Figure 13a,b. It can be seen that the correlation between variable 32, reactor cooling water flow, and variable 21, reactor cooling water outlet temperature, is totally opposite. It is obvious that the reactor cooling water outlet temperature is caused by reactor cooling water flow in normal operation conditions. However, variable correlation shows opposite results in two different abnormal operations.

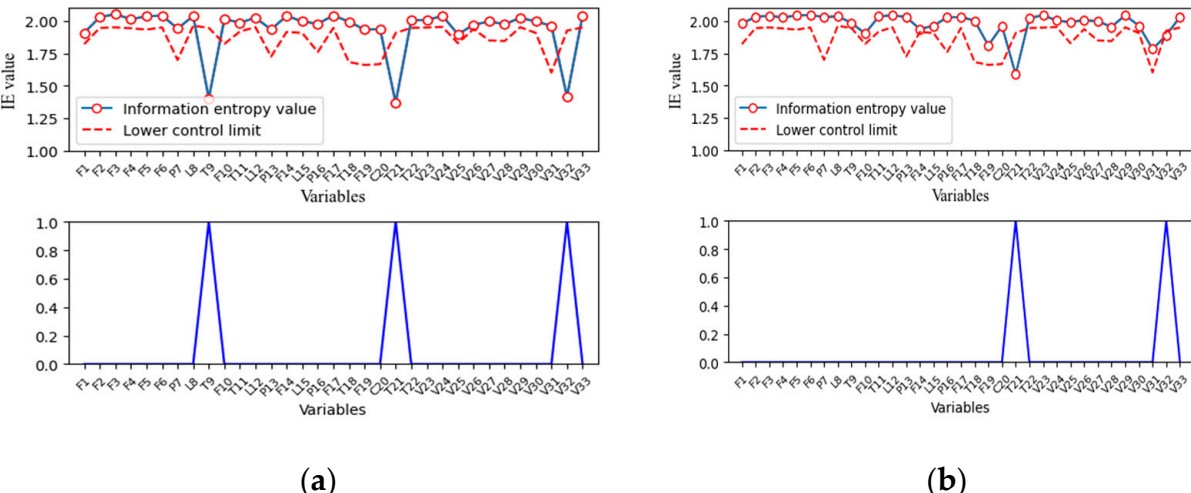

**Figure 12.** (**a**) IE value of each variable in fault 14; (**b**) IE value of each variable in fault 17.

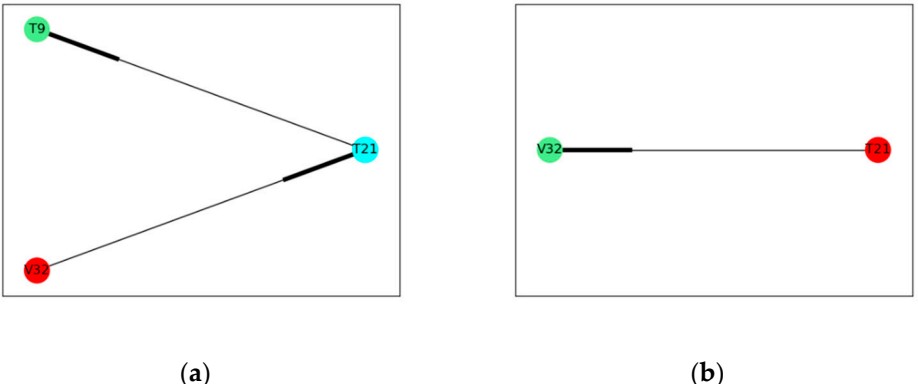

(**a**)          (**b**)

**Figure 13.** (**a**) Fault propagation digraph of fault 14; (**b**) fault propagation digraph of fault 17.

In fault 14, the fault is introduced to the reactor cooling water valve. The reactor cooling water flow reacts the most quickly to the fault and then influences the reactor cooling water outlet temperature. The change of reactor cooling water temperature finally affects the reactor temperature. The fault propagation is V32→T21→T9, which is consistent with the fault description. In fault 17, the fault propagation is from reactor cooling water outlet temperature to reactor cooling water flow. It can be concluded that the fault is introduced to the reactor cooling water outlet temperature, and the reactor cooling water valve responds to the control system to maintain a stable reactor temperature. This fault is defined as unknown fault, but the result can be verified according to data or literature.

The influence of control systems on the correlation among process variables can be further proved through comparing fault diagnosis results of fault 11 and fault 14. In fault 11, the fault is introduced to reactor cooling water inlet temperature, which is not measured. In most literature, the root cause of this fault is diagnosed as variable 32, reactor cooling water flow, but its true reason is the variation of the temperature variable in the reactor. The variation of the reactor cooling water flow is just a response to the control system. The adjustment of the reactor cooling water flow can only mitigate the change of reactor temperature temporarily, the root of this fault is still not removed. Therefore, the reactor cooling water inlet temperature continues to vary randomly, and the system still operates in an abnormal condition. As shown in Table 3, the root cause is diagnosed as the reactor temperature and the reactor cooling water flow in this work, but the time lag between these two variables is zero. The fault propagation cannot be obtained because the reactor temperature is controlled by the reactor cooling water flow. When the reactor temperature varies, the reactor cooling water flow responds quickly to mitigate this change, and the

sample frequency in TEP is 3 min, which is much longer than the response time of the control system.

This point of view can be proved by analyzing fault 14. The root cause of fault 14 is the sticking fault in the reactor cooling water flow, the fault is propagated to the reactor temperature using a few minutes according to the fault propagation results in Figure 13a. It takes a certain time for changes in the reactor cooling water flow to affect the reactor temperature because of fluid flow and heat exchange. However, when an abnormal deviation occurs at the reactor temperature, the control system responds immediately to the reactor cooling water valve. The time lag between the reactor temperature and the reactor cooling water flow in fault 11 is zero; therefore, the root cause of fault 11 is the reactor temperature rather than the reactor cooling water flow. It should be noticed that, in practical situations, the reactor temperature cannot be the root cause as a state variable; then, the fault should be located in the cooling water, and, because the reactor cooling water flow is not the root cause, the real root cause of this fault is the cooling water inlet temperature. Because this variable is not measured in this system, the fault is located in the reactor temperature.

According to literature, a process knowledge based fault diagnosis method and a traditional data based method are referred to compare the results with the proposed method. In Peng Di's study, the fault diagnosis method is implemented by a probabilistic signed directed graph model constructed from the process flowchart of TEP. The results are shown in Figure 14. It can be observed that the reactor cooling water outlet temperature ($T_{rc}$) is caused by reactor cooling water flow ($V_{10}$). However, variable correlation is completely different in a causal model among process variables constructed by a Bayesian network using historical data in Sylvain Verron's study. Their results are shown in Figure 15, where the change of reactor cooling water flow (X51) is caused by the disturbance of reactor cooling water outlet temperature (X21) [29]. It can be seen that faults 14 and 17 cannot be correctly diagnosed by only one method, as causal relations among variables may change significantly due to different control strategies and operating conditions. By the proposed fault diagnosis method, real-time data are applied to calculate the variable correlation, and, therefore, fault diagnosis results obtained from the proposed method are more objective than previous methods.

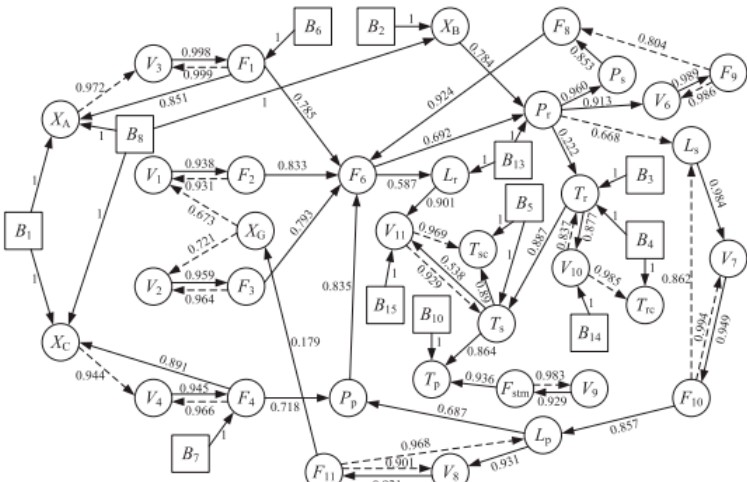

**Figure 14.** Probabilistic signed digraph of the TE process in Peng Di's study. Reproduced with permission from [Peng, D.; Gu, X.; Xu, Y.; Zhu, Q.], [Journal of Loss Prevention in the Process Industries]; published by [Elsevier], 2015.

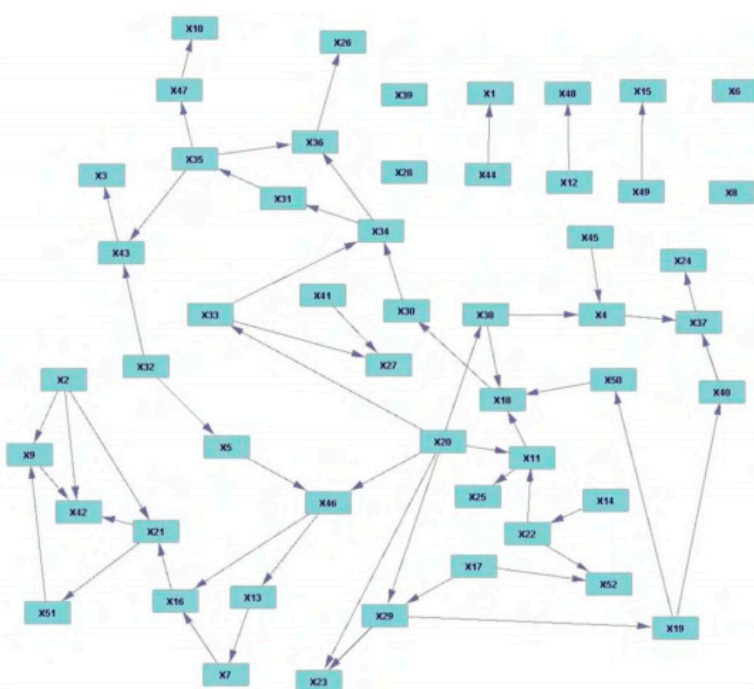

**Figure 15.** Causal Bayesian network of the TEP in Sylvain Verron's study. Reproduced with permission from [Verron, S.; Li, J.; Tiplica, T.], [Journal of Process Control]; published by [Elsevier], 2010.

*4.2. Fault Propagation Analysis in Continuous Catalytic Reforming Facility*

In this section, the proposed fault diagnosis method is applied to a continuous catalytic reforming facility. There are four reactors, four heating furnaces, and a plate heat exchanger in this facility. Among these units, the heat exchanger is of great importance because its performance directly affects the load of reforming reaction furnace and reactor, which has a great influence on energy consumption. In normal operating conditions, pressure difference at hot side is controlled at a stable level. Generally, the increase of pressure difference caused by the adjustment of production loads is also considered as normal operation because the pressure difference is influenced by a naphtha feed flow. However, it is considered as abnormal deviation if the pressure difference increases when the naphtha feed flow is steady. The abnormal increase of pressure difference has a great influence on this system. Due to the abnormal increase in the pressure drop at the hot side, the temperature difference at the hot end will increase, and the heat exchange efficiency of the plate heat exchanger will be compromised, resulting in a substantial increase in the load of the heating furnace and fuel gas consumption. Therefore, it is important to identify the increase of pressure drop early and diagnose the root cause.

The proposed fault diagnosis method is applied to industrial data within the entire year of this facility. Once an abnormal increase in the pressure drop is detected, a root cause can be corrected diagnosed by the proposed method, which can be verified by process operating staff. According to the fault diagnosis results, the abnormal increase of pressure drop is mainly caused by two factors. Next, two parts of data are discussed corresponding to these two factors. Variables related to the above unit in the continuous reforming reaction facility are shown in Table 4, and data in normal operating conditions are selected as training data in the proposed method to obtain the thresholds. The sampling period of data is 1 min. Once the fault is detected, IE and MI in test data are calculated and compared with thresholds obtained previously, the fault propagation path is then obtained.

**Table 4.** Process variables in the continuous catalytic reforming unit.

| Variable | Description | Variable | Description |
|----------|-------------|----------|-------------|
| T01 | Outlet temperature at cold side | PD15 | Reactor pressure drop 3 |
| T02 | Inlet temperature at hot side | PD16 | Reactor pressure drop 4 |
| T03 | Inlet temperature at cold side | T17 | Furnace outlet temperature 1 |
| T04 | Outlet temperature at hot side | T18 | Reactor outlet temperature 1 |
| F05 | Naphtha feed flow | T19 | Furnace outlet temperature 2 |
| F06 | Circulating hydrogen flow | T20 | Reactor outlet temperature 2 |
| PD07 | Inlet filter pressure drop at cold side | T21 | Furnace outlet temperature 3 |
| PD08 | Inlet pressure at cold side | T22 | Reactor outlet temperature 3 |
| P09 | Circulating hydrogen pressure | T23 | Furnace outlet temperature 4 |
| PD10 | Outlet pressure at cold side | T24 | Furnace temperature drop 1 |
| PD11 | Pressure drop at hot side | T25 | Furnace temperature drop 2 |
| PD12 | Pressure drop at cold side | T26 | Furnace temperature drop 3 |
| PD13 | Reactor pressure drop 1 | T27 | Furnace temperature drop 4 |
| PD14 | Reactor pressure drop 2 | PD28 | Reactor inlet pressure |

In case one, the proposed method responds once this abnormal condition is detected, and fault diagnosis results are shown in Figure 16. It can be concluded that this abnormal increase in pressure drop at the hot side is caused by the circulating hydrogen flow. The result can be proved by the original data shown in Figure 17. It can be obtained that the circulating hydrogen flow increases rapidly at the 380th sample, and then the pressure difference at the hot side increases accordingly. By further analyzing with engineering experience, the circulating hydrogen flow is adjusted with changes of environmental temperature because the concentration of circulating hydrogen is influenced by temperature. The environmental temperature and the concentration of circulating hydrogen cannot be measured in real time by DCS. Therefore, the circulating hydrogen flow is diagnosed as the root cause, which is reasonable and instructive to operators.

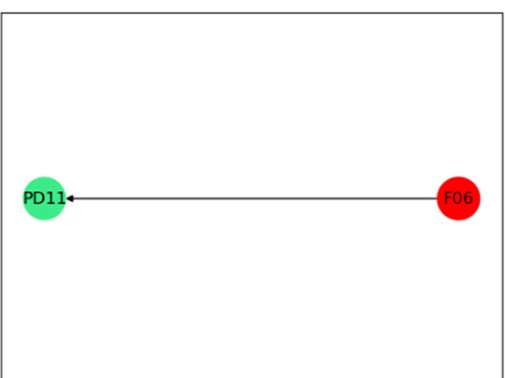

**Figure 16.** Fault propagation digraph in case one.

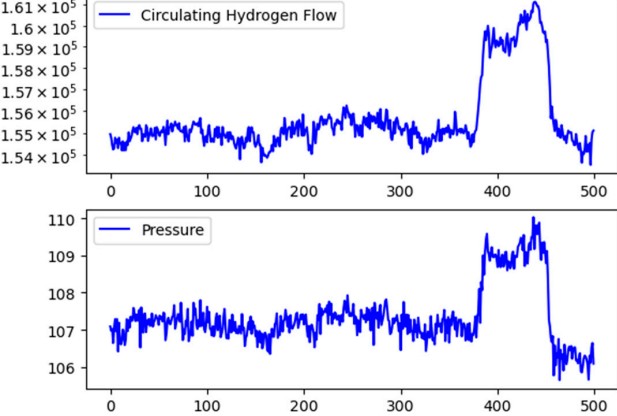

**Figure 17.** Original data in case one.

In case two, an abnormal increase of pressure drop is detected again. However, the circulating hydrogen flow stays steady during this time. The fault diagnosis result obtained from the proposed method is shown in Figure 18. It can be concluded that this abnormal increase of pressure drop is caused by inlet temperature at cold side, and then outlet temperature at hot side is further influenced. According to original data in Figure 19, the pressure difference drop at hot side increases at the 350th sample, but inlet temperature at cold side increases as early as the 67th sample, indicating that the fault diagnosis result is correct. The results show that the fault propagation path and the root cause can be identified by the proposed method correctly, and the proposed method can be well applied to industrial processes.

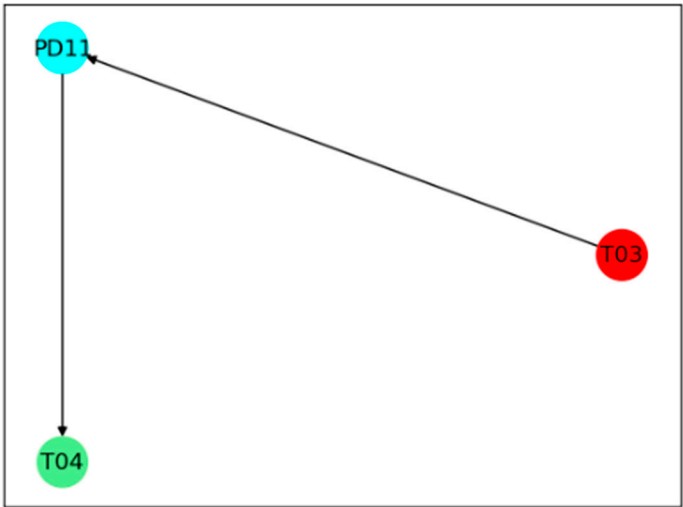

**Figure 18.** Fault propagation digraph in case two.

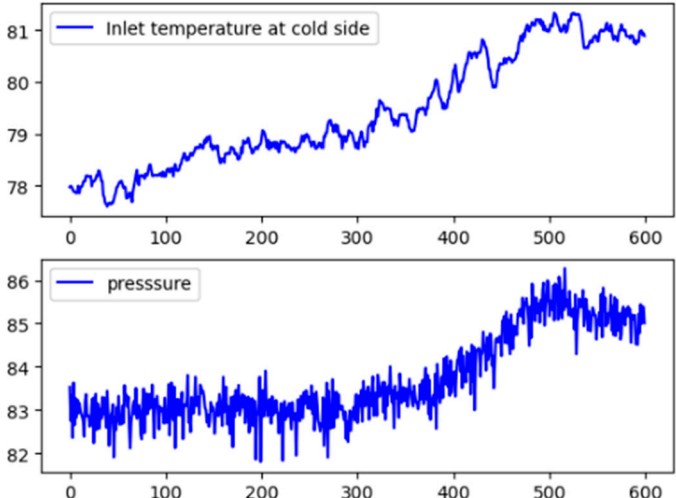

**Figure 19.** Original data in case two.

## 5. Conclusions

In this work, variable correlation under normal operating conditions and faulty states is analyzed by MI. It can be concluded that variable correlation is mostly contributed from random noises under normal conditions. Once a faulty condition is detected, the true correlation can be identified if the fault is propagated between any two variables. Therefore, a fault diagnosis framework is proposed for fault propagation analysis and root cause identification. In steady state, correlation between each pair of variables caused by random noises is calculated to obtain a threshold. The fault propagation path is identified

by TDMI between process variables using real-time data. The proposed method is applied to a simulated example, TEP, and an industrial process. The results show that not only can the fault propagation path be reasonably identified, but the root cause of faults can also be effectively isolated. It is worth noting that variable correlation under normal operation conditions can be different with that under faulty conditions in certain circumstances. The difference in variable correlation under different operation conditions can be effectively captured by the proposed method, which provides a more objective way to identify the fault propagation than previous fault diagnosis methods based on causal relations established by process knowledge or process data only under normal operating conditions. The logic of this fault diagnosis method can be applied to industrial processes with multiple operating conditions and complex control loops because variable correlation is solely captured from online data.

However, it is worth noticing that the calculation of TDMI is conducted by time-delayed windows among pair variables, by which the causal relations among them are determined. A certain propagation path may not be able to be identified if the sampling rate is slower than its propagation dynamic. In that case, both fast sampling rate and operator knowledge can help by proper integration of the proposed method in future study.

**Author Contributions:** Conceptualization, W.S.; Methodology, C.J.; Software, C.J. and F.M.; Validation, C.J. and W.S.; Formal Analysis, C.J.; Investigation, C.J. and F.M.; Resources, W.S. and J.W. (Jingde Wang); Data Curation, W.S. and C.J.; Writing—Original Draft Preparation, C.J.; Writing—Review and Editing, W.S., J.W. (Jianhong Wang) and C.J.; Visualization, C.J.; Supervision, W.S. and J.W. (Jianhong Wang); Project Administration, W.S. and J.W. (Jingde Wang); Funding Acquisition, W.S. All authors have read and agreed to the published version of the manuscript.

**Funding:** This research was funded by the National Natural Science Foundation of China (Grant No. 21878012).

**Institutional Review Board Statement:** Not applicable.

**Informed Consent Statement:** Not applicable.

**Data Availability Statement:** Not applicable.

**Conflicts of Interest:** The authors declare no conflict of interest. The funders had no role in the design of the study, in the collection, analyses, or interpretation of data, in the writing of the manuscript, or in the decision to publish the results.

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
