# Peer review of "Real-Time Industrial Process Fault Diagnosis Based on Time Delayed Mutual Information Analysis"

_processes, doi:10.3390/pr9061027_

Round 1

Reviewer 1 Report

The work presented in the paper is commendable and the scientific level is suitable for the journal. I only want to suggest some minor improvements:

- a lot of abbreviations are used within the paper (DCS, IE, MI, TEP, TDMI), which are explained when introduced. Even if that approach is correct, it forces the reader to go back and forth between sections as he reads the paper, which worsens the understanding of its contents. I suggest that the authors introduce a short paragraph at the end of the introductory section outlining all the abbreviations used in the paper

- the conclusions section is rather short (the usefulness of the proposed approach should be presented in more detail) and no further research directions are introduced

Reviewer 2 Report

The presented paper describes algorithm to calculate the fault diagnosis. The work is interesting and well written. However, the fault diagnosis has a lot of known methods. From my point of view, the work must have comparison with different algorithm showing the benefits of the new one. The quantitive and qualitive comparison should be performed.

The minor comments:

1) Figure 9. Fault propagation digraph result. - the figure is very trivial

2) Can we find the type of fault based on this research.

Round 2

Reviewer 2 Report

The work was updated according to remarks.